

# Genomic data mining of the marine actinobacteria *Streptomyces* sp. H-KF8 unveils insights into multi-stress related genes and metabolic pathways involved in antimicrobial synthesis

Agustina Undabarrena[1], Juan A. Ugalde[2], Michael Seeger[1] and Beatriz Cámara[1]

[1] Departmento de Química & Centro de Biotecnología, Universidad Técnica Federico Santa María, Valparaiso, Chile

[2] Centro de Genética y Genómica, Facultad de Medicina Clinica Alemana, Universidad del Desarrollo, Santiago, Chile

## ABSTRACT

*Streptomyces* sp. H-KF8 is an actinobacterial strain isolated from marine sediments of a Chilean Patagonian fjord. Morphological characterization together with antibacterial activity was assessed in various culture media, revealing a carbon-source dependent activity mainly against Gram-positive bacteria (*S. aureus* and *L. monocytogenes*). Genome mining of this antibacterial-producing bacterium revealed the presence of 26 biosynthetic gene clusters (BGCs) for secondary metabolites, where among them, 81% have low similarities with known BGCs. In addition, a genomic search in *Streptomyces* sp. H-KF8 unveiled the presence of a wide variety of genetic determinants related to heavy metal resistance (49 genes), oxidative stress (69 genes) and antibiotic resistance (97 genes). This study revealed that the marine-derived *Streptomyces* sp. H-KF8 bacterium has the capability to tolerate a diverse set of heavy metals such as copper, cobalt, mercury, chromate and nickel; as well as the highly toxic tellurite, a feature first time described for *Streptomyces*. In addition, *Streptomyces* sp. H-KF8 possesses a major resistance towards oxidative stress, in comparison to the soil reference strain *Streptomyces violaceoruber* A3(2). Moreover, *Streptomyces* sp. H-KF8 showed resistance to 88% of the antibiotics tested, indicating overall, a strong response to several abiotic stressors. The combination of these biological traits confirms the metabolic versatility of *Streptomyces* sp. H-KF8, a genetically well-prepared microorganism with the ability to confront the dynamics of the fjord-unique marine environment.

Corresponding author
Beatriz Cámara,
beatriz.camara@usm.cl

## INTRODUCTION

There has been a burst of genomic data in recent years due to the advances in various technologies such as next-generation sequencing. Whole genome sequencing is providing information-rich data that can hugely contribute and orientate the discovery of natural products in microorganisms. Indeed genome mining has been positioned as a fundamental

bioinformatics-approach in the natural product field (*McAlpine et al., 2005*; *Van Lanen & Shen, 2006*; *Challis, 2008*; *Doroghazi & Metcalf, 2013*; *Jensen et al., 2014*; *Antoraz et al., 2015*; *Tang et al., 2015a*; *Tang et al., 2015b*; *Katz & Baltz, 2016*). Natural products have clearly demonstrated to play a significant role in drug discovery, in fact 78% of antibiotics marketed during 1982–2002 originated from natural products (*Peláez, 2006*). Considering the year 2014, 25% of the approved new chemical entities were from natural or natural-derived products (*Newman & Cragg, 2016*). In natural environments, these metabolites also play important roles as signal molecules, facilitating intra- or inter-species interactions within microbial communities related to virulence, colonization, motility, stress response and biofilm formation (*Romero et al., 2012*).

*Streptomyces* are mycelium-forming bacteria with a complex developmental life cycle that includes sporulation and programmed cell death processes (*Flärdh & Buttner, 2009*; *Yagüe et al., 2013*). Their unsurpassed richness and diversity concerning secondary metabolism pathways has made them valuable providers for bioactive molecules, being responsible for two-thirds of all known antibiotics (*Bérdy, 2012*). Genome mining has become a powerful tool to unveil the biotechnological potential of *Streptomyces* species, where biosynthetic gene clusters (BGCs) can be identified (*Weber et al., 2015*) and even predict the chemical core structure of the molecules. Unlike other bacteria, *Streptomyces* have linear chromosomes (*Chen et al., 2002*) and their genome sizes are within the largest in the bacterial world (*Weber et al., 2003*), ranging from 6.2 Mb for *Streptomyces cattleya* NRRL 8057 (*Barbe et al., 2011*) to 12.7 Mb for *Streptomyces rapamycinicus* NRRL 5491 (*Baranasic et al., 2013*), considering complete sequenced genomes to date (*Kim et al., 2015*). Up to 5% of their genomes are devoted to the synthesis of secondary metabolites (*Ikeda et al., 2003*). The ability to produce a wide variety of bioactive molecules is based on the fact that they contain the largest numbers of BGCs such as polyketide synthases (PKS) and non-ribosomal peptide synthetases (NRPS), or even PKS-NRPS hybrids (*Challis, 2008*). The genes required for secondary metabolites biosynthesis are typically clustered together (*Zazopoulos et al., 2003*) and are tightly regulated both by specific regulation of each product (*Bibb & Hesketh, 2009*) or by pleiotropic mechanisms of regulation that can control several pathways at the same time (*Martin & Liras, 2012*). Due to these interesting properties, nearly 600 species and 30,000 strains of *Streptomyces* have been identified (*Euzéby, 2011*). To date, 653 *Streptomyces* genome assemblies are available in the GenBank database (*Studholme, 2016*) and this number is likely to keep increasing.

Although soil microorganisms from the *Streptomyces* genus have generated vast interest due to their exceptional role as antibiotic producers (*Bérdy, 2012*), their marine counterpart has been less explored. The marine ecosystem is highly diverse, with extreme abiotic selective pressures and immense biological diversity (*Lam, 2006*). In addition, many marine organisms have a sessile life style, needing chemical weapons for defense and survival (*Haefner, 2003*). Thus, research in natural products has been focusing on the isolation of microorganisms from corals (*Hodges, Slattery & Olson, 2012*; *Kuang et al., 2015*; *Mahmoud & Kalendar, 2016*; *Pham et al., 2016*), sponges (*Kim, Garson & Fuerst, 2005*; *Montalvo et al., 2005*; *Zhang et al., 2006*; *Jiang et al., 2007*; *Vicente et al., 2013*; *Sun et al., 2015*), as well as marine sediments (*Mincer & Jensen, 2002*; *Magarvey et al., 2004*;

*Jensen et al., 2005*; *León et al., 2007*; *Gontang, Fenical & Jensen, 2007*; *Duncan et al., 2014*; *Yuan et al., 2014*). In spite of all the isolation studies associated to marine actinobacteria, relatively little is known about the molecular mechanisms behind bacterial adaptation to marine environments. It is supposed that marine actinobacteria have adapted through the development of specific biological traits (*Tian et al., 2016*), which has led to hypothesize that novel species from unexplored habitats may contain unique bioactive compounds (*Axenov-Gribanov et al., 2016*). In addition, marine habitats are under a dramatic pollution increase, where heavy metals have demonstrated to be one of the most negative causing impacts in living beings. While many metals (iron, zinc, manganese, copper, cobalt, nickel, vanadium, molybdenum) are essential micronutrients for enzymes and cofactors, they still are toxic when available in high concentrations, causing adversary effects mainly by oxidative stress damage to fundamental macromolecules (*Schmidt et al., 2005*). In this context, marine microorganisms have developed mechanisms through molecular adaptations in order to thrive in these adverse conditions. Moreover, secondary metabolites biosynthesis are strongly influenced by the presence and concentration of certain heavy metals in *Streptomyces* genus (*Locatelli, Goo & Ulanova, 2016*), and also oxidative stress can regulate antibiotic production (*Kim et al., 2012*; *Beites et al., 2014*) providing evidence of a molecular crosstalk response between these stressors.

In the South Pacific region, Chile has an extensive marine coast that remains mostly unexplored. Bioprospecting of actinobacteria for the discovery of novel marine-derived natural products, specifically antibiotics, has been carried out in Valparaíso Central Bay (*Claverías et al., 2015*) and in the Comau fjord in Northern Patagonia (*Undabarrena et al., 2016*). Both sites proved to be a rich source for novel species of actinobacteria with antimicrobial properties. In this context, the genome of a selected antimicrobial-producer marine *Streptomyces* strain from Comau fjord was sequenced (*Undabarrena et al., 2017*). In this study, we aimed to conduct a combined genomic, metabolic and physiological analysis of the marine *Streptomyces* sp. H-KF8 bacterium, through the further exploration of its antimicrobial activity and the genome mining of the BGCs encoded in its genome. In addition, the genetic and functional response to abiotic stressors such as oxidative stress, heavy metals and antibiotics, which may play an important role in the evolution of secondary metabolism genes, was evaluated in *Streptomyces* sp. H-KF8.

## METHODS AND MATERIALS

### Bacterium selection

Underwater samples were previously collected from marine sediments from the Marine Protected Area of the Comau fjord, in the Northern Chilean Patagonia (*Undabarrena et al., 2016*). Fjords are especially attractive due to its unique biogeographic characteristics, being a deep narrow inlet with significantly eroded bottom and communication with open sea (*Bredhold et al., 2007*). Comau fjord is one of the deepest; it has a high precipitation rate crucial for fresh water input; where water surface temperatures ranges between 5 °C and >20 °C, sustaining a thermohaline circulation (*Bustamante, 2009*; *Sobarzo, 2009*). As microorganisms of these ecosystems may display remarkable genetic features of tolerance

to the dynamics of these abiotic stressors, marine actinobacteria were isolated with several culture media and identified through 16S rRNA gene sequence (*Undabarrena et al., 2016*). Antimicrobial potential was screened using two strategies, including assessing the antimicrobial activity of crude extracts derived from liquid cultures (*Undabarrena et al., 2016*). *Streptomyces* sp. H-KF8 was selected due to its antimicrobial activities against *S. aureus, L. monocytogenes* and *E. coli* for whole genome sequencing, representing the first genome of Chilean marine actinobacteria (*Undabarrena et al., 2017*).

## Phenotypic characterization

*Streptomyces* sp. H-KF8 was characterized morphologically in several media agar plates: ISP1-ISP9 (*Shirling & Gottlieb, 1966*), Marine Agar (MA) 2216 (Difco) and Triptic Soy Agar (TSA) (Difco NO 236950). All media, with exception of MA, were prepared with artificial sea water (ASW) (*Kester et al., 1967*) as the strain has a specific ASW requirement for growth (*Undabarrena et al., 2016*; *Undabarrena et al., 2017*). Plates were incubated at 30 °C and visible colonies appeared after 5–7 days. Microscopic images were obtained with a Leica Zoom2000 stereoscope (Arquimed), Gram-staining was performed with an optical microscope L2000A (Arquimed) with 1,000× magnification, and unstained low voltage electron microscopy (LVEM) was used for high contrast images (Delong LVEM5 microscope, Universidad Andrés Bello, Chile) after 21 days of *Streptomyces* sp. H-KF8 growth in ISP3-ASW media (*Vilos et al., 2013*).

## Antimicrobial activity

Antimicrobial activity was evaluated previously in ISP2 and TSA-ASW agar plates, and activity was corroborated by liquid culture derived crude extracts (*Undabarrena et al., 2016*). In this study, a further evaluation of antimicrobial activity was assessed in 15 different media agar plates, to explore the relation between *Streptomyces* sp. H.KF8 morphology and antimicrobial activity. Various media were employed: ISP1-ISP9; MA; King B; Medium V (*Marcone et al., 2010*); LB-ASW; Actino Agar (Difco) and NaST21Cx (*Magarvey et al., 2004*), using cross-streak method as previously described (*Haber & Ilan, 2014*). The assay was slightly modified for marine actinobacteria by our group (*Claverías et al., 2015*; *Undabarrena et al., 2016*). Antimicrobial activity was measured against five reference bacteria: *Staphylococcus aureus* NBRC 100910[T]; *Listeria monocytogenes* 07PF0776; *Salmonella enterica* subsp enterica LT2[T]; *Escherichia coli* FAP1 and *Pseudomonas aeruginosa* DSM50071[T]. Briefly, inhibition zones were seen as part of the bacterial line where the reference bacteria did not grow, and ranked qualitatively as: −, no inhibition; ±, attenuated growth of target bacterium; +, <50% growth inhibition of target bacterium (1–5 mm of the line); ++, 50% growth inhibition of target bacterium (6–10 mm of the line); +++, >50% growth inhibition of target bacterium (≥11 mm of the line). All experiments were performed in duplicate, using as internal control one of the reference bacteria.

Additionally, the double-layer method (*Westerdahl et al., 1991*) was employed, in order to perform a time-course assay to ascertain the days of incubation where most activity was being produced. *Streptomyces* sp. H-KF8 macrocolonies were incubated on ISP2-ASW, ISP3-ASW, TSA-ASW and MA agar plates. Macrocolonies were grown individually from

five to 20 days on the same agar plate, and subsequently, 7 mL of modified-LB (7 g/L of agar instead of 15 g/L) with an aliquot of 100 µL of an overnight pre-grown *S. aureus* bacterial culture with an OD = 0.3 was added above the macrocolonies of *Streptomyces* sp. H-KF8. Inhibition zones were observed after incubation of plates for 24 h at 37 °C. If inhibition zones overlapped, the experiment was repeated on separate agar plates, where only one macrocolony in the center of the plate was incubated.

## Genome mining and bioinformatic analysis

*Streptomyces* sp. H-KF8 whole genome sequencing was performed by Illumina and PacBio (*Undabarrena et al., 2017*). Genome reads were *de novo* assembled using Canu (version 1.1) (*Berlin et al., 2015*) into 11 contigs, representing one linear chromosome of 7,684,888 bp genome. Full genome sequencing details can be found elsewhere (*Undabarrena et al., 2017*). Gene calling an annotation was performed using the Prokaryotic Genome Annotation Pipeline (PGAP) at NCBI (version 3.1) (*Tatusova et al., 2016*). Genes were assigned to EggNOG categories (*Huerta-Cepas et al., 2016*) via an HMM search with HMMER3 (http://hmmer.org). Genetic determinants involved in biological traits analyzed in this report were manually established and the amino acidic signatures were validated based on domain hits through Basic Local Alignment Search Tool (BLAST) from NCBI. Also, BGCs were identified through AntiSMASH (version 3.0) online platform. Snapgene software (version 2.3.2) was used to visualize ORFs related to functional biological traits from each linear contig. Artemis software (version 16.0.0) was used to construct the graphic representation of the circular chromosome, and to assign by colors manually all the different categories of BGCs on it.

## Functional response to heavy metal(loid)s

For metal-resistance experiments, agar plates containing filtered salts of several metal(loid) solutions were prepared. Metals were diluted to obtain the following final concentration in media plates: $CuSO_4$ (0.25 mM, 0.5 mM and 0.75 mM); $CoCl_2$ (2 mM, 4 mM and 6 mM); $ZnSO_4$ (50 mM and 100 mM); $CdCl_2$ (0.75 mM and 1.5 mM); $HgCl_2$ (20 µM, 40 µM and 60 µM); $K_2TeO_3$ (10 µM, 20 µM and 40 µM); $K_2CrO_4$ (10 µM, 17 µM and 20 µM); $Na_2HAsO_4$ (50 mM and 100 mM); $NaAsO_3$ (2.5 mM and 5 mM) and $NiSO_4$ (5 mM, 10 mM and 15 mM). *Streptomyces* sp. H-KF8 was evaluated after 5, 10 and 20 days of growth in TSA-ASW plates. Additionally, a special Minimal Medium (MM) used to evaluate metal resistance in *Streptomyces* spp. was prepared (*Schmidt et al., 2009*), modified with the addition of ASW. Experiment was performed with two biological replicates. Reference values for metal concentrations were decided based on metal-tolerance *Streptomyces* obtained from literature (*Schmidt et al., 2005*; *Schmidt et al., 2009*; *Wang et al., 2006*; *Polti, Amoroso & Abate, 2007*). Agar plates without addition of any metals were prepared as negative controls.

## Functional response to oxidative stress

For oxidative stress experiments, tolerance to hydrogen peroxide ($H_2O_2$) at various concentrations (0.2 M, 0.5 M, 1 M, 2 M, and 4 M) was evaluated by directly adding 10 µL of the $H_2O_2$ solution to a sterile paper disk positioned on a TSA-ASW agar plate where

*Streptomyces* sp. H-KF8 was streaked out to grow as a thin lawn (*Dela Cruz et al., 2010*). The model strain *Streptomyces violaceoruber* A3(2) (DSM 40783) was used to test the tolerance response. Inhibitions areas ($cm^2$) were observed after 5 days of growth at 30 °C. Experiment was performed with three biological replicates, and standard deviation was calculated. A statistical analysis by Student's *t*-Test was carried out considering a *p*-value <0.01.

### Functional response antibiotics

Susceptibility to model antibiotics of *Streptomyces* sp. H-KF8 was explored previously (*Undabarrena et al., 2016*). However, in this report a further characterization was pursued. *Streptomyces* sp. H-KF8 was grown on Mueller-Hinton agar plates prepared with ASW (MH-ASW) and commercial standard disks of model antibiotics were placed above. The following antibiotics were tested: Amoxicillin 25 µM, Bacitracin 0.09 IU, Novobiocin 5 µg and Erythromycin 15 µg (LabClín); Optochin 5 µg (BritaniaLab); Clindamycin 2 µg, Oxacillin 1 µg, Ciprofloxacin 5 µg, Ceftriaxone 30 µg, Chloramphenicol 30 µg, Penicillin 10 UOF, Cefotaxime 30 µg, Gentamicin 10 µg and Ampicillin 10 µg (Valtek). After 5 days of incubation at 30 °C, radios of the inhibition halos were measured, and inhibition areas ($cm^2$) were calculated. Data was compared with standarized cut off values from Clinical and Laboratory Standards Institute (CLSI) from year 2016, to determine susceptibility or resistance against each antibiotic tested. Experiments were performed using three biological replicates, and standard deviation was calculated for each antibiotic.

## RESULTS

### Phenotypic characterization

Morphological analysis of *Streptomyces* sp. H-KF8 was carried out by strain growth in several media, containing different carbon sources (Fig. 1; inset colony morphology). Growth of *Streptomyces* sp. H-KF8 was observed in the standard ISP1-ISP9 agar plates, although differences in growth rates and pigmentation were noticed (Figs. 1A–1F). On ISP1 (yeast extract, pancreatic digest of casein), ISP2 (yeast extract, malt extract, dextrose) and ISP6 (peptone, yeast extract and iron) media, white mycelia was observed, with appearance of grayish-spores after 14 days of growth. In contrast, when *Streptomyces* sp. H-KF8 was grown on ISP3 (outmeal), ISP4 (soluble starch and inorganic salts), ISP5 (glycerol and asparagine) and ISP9 (glucose) media, creamy mycelia was observed, with appearance of white spores at the periphery of the colonies. In contrast, poor growth was observed in ISP7 (tyrosine) medium. A different morphology was perceived when *Streptomyces* sp. H-KF8 was grown on MA medium (Fig. 1G). Colony size was comparatively smaller (5.06 ± 1.1 mm in ISP2 vs 3.12 ± 0.78mm in MA; *p* < 0.01), and a dark-grey turning into black pigmentation was noticed within 5 days of growth. On TSA-ASW plates, a white mycelium was observed with no change in pigmentation over time, but with presence of exudate drops in the colony surface (Fig. 1H). Additionally, morphology was visualized microscopically, and typical *Streptomyces* structures of development such as hyphae and spores were observed (Fig. 2). Exudate drops were appreciated in ISP2 medium during late growth phase (Fig. 2A), spores were identified with optical microscopy (Fig. 2B) and

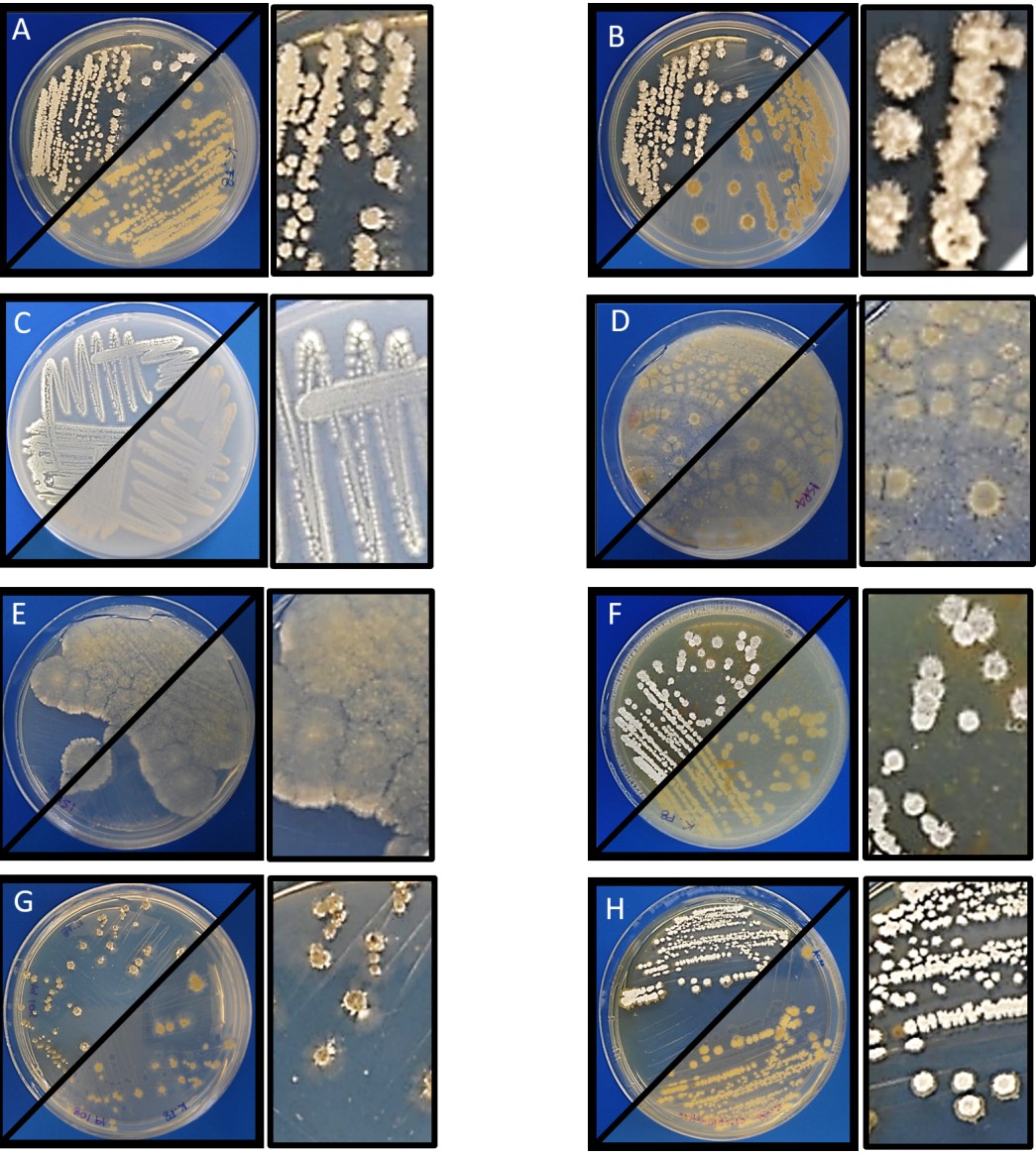

**Figure 1** **Morphology of *Streptomyces* sp. H-KF8.** Macrocolony showing anverse and reverse growth in several media. Inset shows a zoom of colony morphology. (A) ISP1-ASW; (B) ISP2-ASW; (C) ISP3-ASW; (D) ISP4-ASW; (E) ISP5-ASW; (F) ISP6-ASW; (G) Marine Agar (MA); (H) TSA-ASW.

hyphae with Gram staining (Fig. 2C). Moreover, the complex network of intertwined hyphae and early spore chain assemblies was observed by LVEM microscopy, which is a distinctive feature of *Streptomyces* genus (Fig. 2D).

## Antimicrobial activity

Antimicrobial activity of *Streptomyces* sp. H-KF8 was further characterized using agar media with different carbon sources (Table 1). In general, antimicrobial activity was more evident against Gram-positive reference bacteria (*S. aureus* and *L. monocytogenes*), although inhibition against *E. coli* was also observed in most media, which is consistent

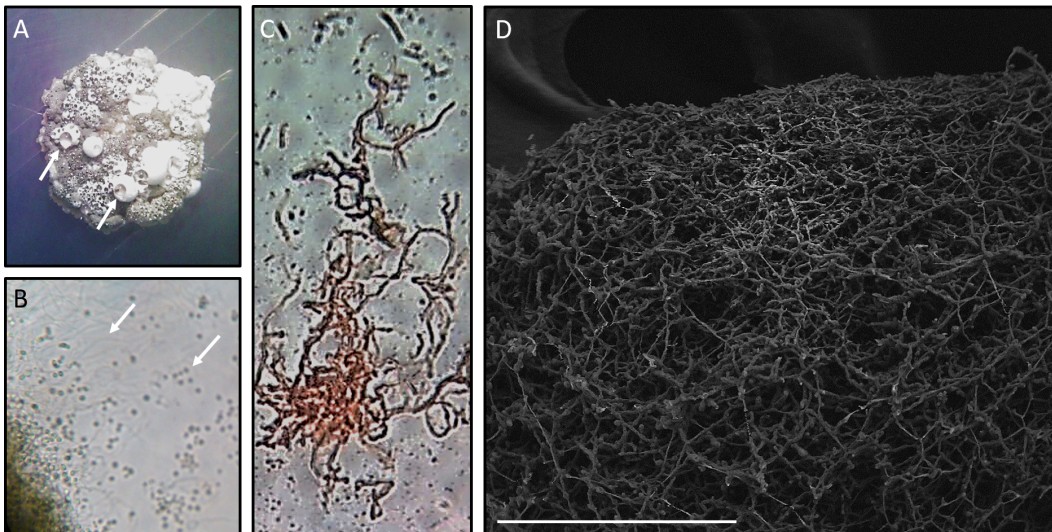

**Figure 2** **Microscopy of *Streptomyces* sp. H-KF8.** (A) Stereoscope zoom of a macrocolony grown in ISP2-ASW agar plate. Arrows shows exudates. (B) Optic Microscopy image at 1,000×. Arrows indicate hyphae and spores, respectively. (C) *Streptomyces* sp. H-KF8 gram staining, showing hyphae. (D) Scanning Electron Microscopy (LVEM) image of *Streptomyces* sp. H-KF8 grown on ISP3-ASW agar plates for 21 days. Bar represent 100 μm.

with results obtained from *Streptomyces* sp. H-KF8 crude extracts (*Undabarrena et al., 2016*). *P. aeruginosa* was the reference bacterium less inhibited. Among the 15 different media used, inhibition of at least one reference bacterium was noted in 87% of the media. Best media for antimicrobial activity were ISP1, ISP2, ISP6, and V media, where inhibition of four of the five reference bacteria was observed. Notably, in ISP2 medium a unique attenuation of *P. aeruginosa* growth was observed. Alternatively, a time-course assay using the double-layer method was performed to visualize the starting day of the antimicrobial activity, in four media that presented inhibition. Even though at day 5 a relatively scarce colony growth of *Streptomyces* sp. H-KF8 was observed in ISP2 medium, at day 6 it was possible to visualize a modest inhibition against *S. aureus* (Fig. 3A). Yet, inhibition zone increased as incubation time for *Streptomyces* sp. H-KF8 extended, as shown in Fig. 3B, showing a maximum halo size at day 15 (Fig. 3C), suggesting a tight relation between growth and antimicrobial activity which is also correlated to the carbon source of the media tested.

## Bioinformatic analysis and genome mining for BGCs

Whole genome sequencing and genome features were previously described (*Undabarrena et al., 2017*). Briefly, *Streptomyces* sp. H-KF8 genome was assembled into 11 contigs, with a total genome length of 7,684,888 bp, and a G + C content of 72.1%. A total of 6,574 genes are represented among 6,486 CDS, 67 tRNAs and 6 16S rRNAs. Genes with coding sequences were grouped into COGs categories, although 808 genes remain ungrouped. Description and gene percentage of each category is depicted in Table 2. For *Streptomyces* sp. H-KF8, the most abundant categories were transcription (522 genes), carbohydrate transport and metabolism (362 genes), and amino acid transport and metabolism (362

**Table 1** Antibacterial activity of *Streptomyces* sp. H-KF8 in several culture media.

| Medium | Bacterial strains[a] | | | | |
| --- | --- | --- | --- | --- | --- |
| | STAU | LIMO | PSAU | SAEN | ESCO |
| ISP1 | +++ | +++ | − | +++ | +++ |
| ISP2 | +++ | +/− | +/− | − | + |
| ISP3 | + | + | − | − | + |
| ISP4 | + | − | − | − | − |
| ISP5 | + | − | − | − | − |
| ISP6 | ++ | +++ | − | + | + |
| ISP7 | +++ | − | − | +/− | ++ |
| ISP9 | +++ | +++ | − | − | − |
| TSA-ASW | +++ | +/− | − | − | + |
| MA | +++ | +++ | − | − | ++ |
| King B | − | +/− | − | − | − |
| Medium V | ++ | ++ | − | +/− | +++ |
| LB-ASW | +++ | ++ | − | +/− | − |
| Actino Agar | − | − | − | − | − |
| NaST21Cx | − | − | − | − | − |

**Notes.**

−, no inhibition; +/−, attenuated growth; +, <50% growth inhibition (1–5 mm ); ++, 50% growth inhibition (6–10 mm); +++, >50% growth inhibition (≥11 mm).

[a]STAU, *S. aureus*; LIMO, *L. monocytogenes*; PSAU, *P. aeruginosa*; SAEN, *S. enterica*; ESCO, *E. coli*.

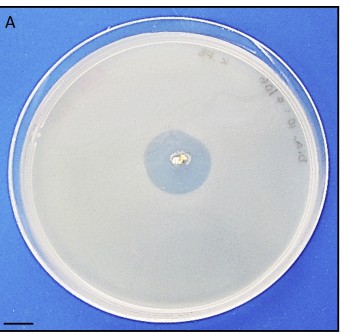
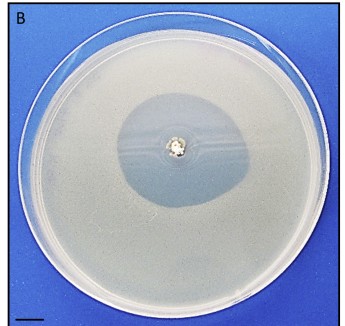
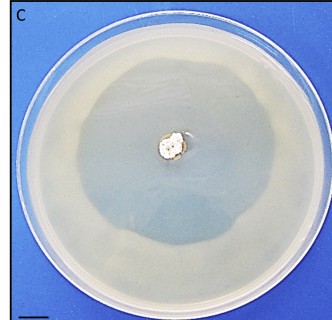

**Figure 3** **Antibacterial activity of *Streptomyces* sp. H-KF8.** Photographs depict inhibition zone against *Staphylococcus aureus*. Bar represents 10 mm. Time course was performed using the double-layer method, at various incubation days: (A) 6 days; (B) 9 days; (C) 15 days.

genes). The *Streptomyces* sp. H-KF8 categorized genes were compared to the model *Streptomyces violaceoruber* A3(2) isolated from soil (*Bentley et al., 2002*) and the marine *Streptomyces* sp. TP-A0598 (*Komaki et al., 2015*), in order to observe if these features could be considered as source-derived traits (Fig. 4). As there are scarce reports available on marine *Streptomyces* genomes that include COGs detailed annotation, *Streptomyces* sp. TP-A0598 is one of the few that have these characteristics, and therefore selected for comparison. While all three strains showed the same tendency in the categories previously named in terms of abundancy, differences were observed in terms of percentage in transcription and carbohydrate metabolism categories, where *S. violaceoruber* A3(2)

**Table 2  COGs distribution of genes with coding sequences in *Streptomyces* sp. H-KF8.**

| COG functional categories | Abbreviation | No of genes | Percentage (%) |
|---|---|---|---|
| Energy production and conversion | C | 275 | 4.18 |
| Cell division and chromosome partitioning | D | 41 | 0.62 |
| Amino acid transport and metabolism | E | 322 | 4.90 |
| Nucleotide transport and metabolism | F | 89 | 1.35 |
| Carbohydrate transport and metabolism | G | 362 | 5.51 |
| Coenzyme transport and metabolism | H | 136 | 2.07 |
| Lipid metabolism | I | 142 | 2.16 |
| Translation | J | 168 | 2.56 |
| Transcription | K | 522 | 7.94 |
| DNA replication and repair | L | 217 | 3.30 |
| Cell envelope biogenesis, outer membrane | M | 169 | 2.57 |
| Cell motility | N | 0 | 0.00 |
| Post-translational modification, protein turnover, chaperones | O | 135 | 2.05 |
| Inorganic ion transport and metabolism | P | 223 | 3.39 |
| Secondary metabolism | Q | 148 | 2.25 |
| General function prediction only | R | 238 | 3.62 |
| Function unknown | S | 2,111 | 32.11 |
| Signal transduction | T | 283 | 4.30 |
| Defense mechanisms | V | 185 | 2.81 |
| Not in COGs | − | 808 | 12.29 |

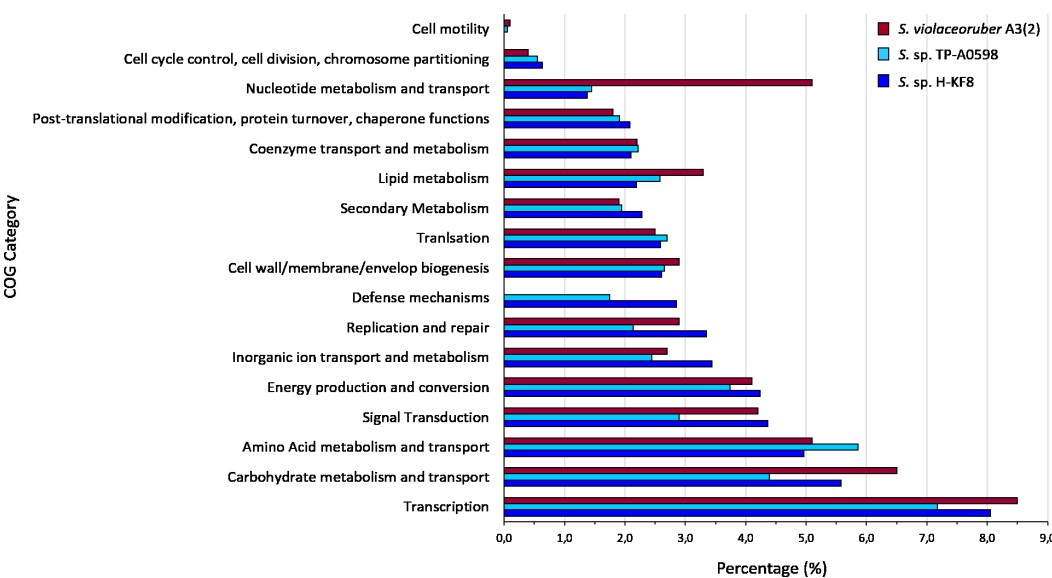

**Figure 4  Comparative genomics of COGs categories.** Percentage of each COG category is shown for the different *Streptomyces* species, where blue is *Streptomyces* sp. H-KF8; light blue is the marine-derived *Streptomyces* sp. TP-A0598; and red is the soil-derived *Streptomyces violaceoruber* A3(2).

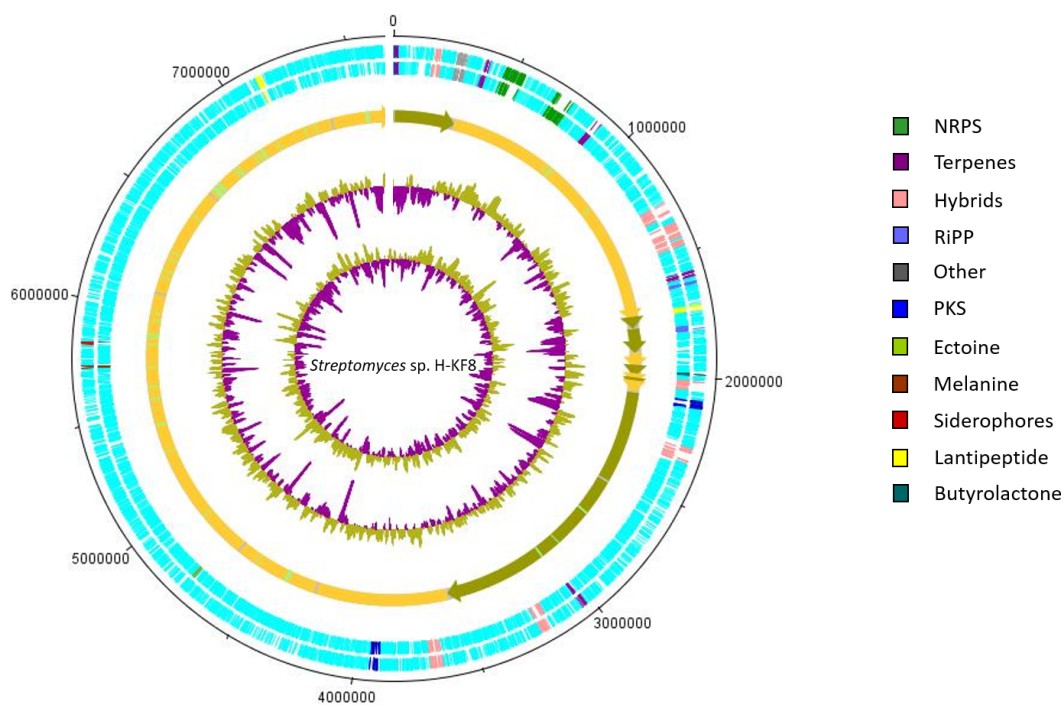

**Figure 5** **Representation of chromosome features and BGCs of *Streptomyces* sp. H-KF8.** Colors depict the different classification types of secondary metabolism gene clusters along the sequenced genome. NRPS, Non-ribosomal peptide synthetase; PKS, polyketide synthase; RiPP, ribosomally synthesized and post-translationally modified peptides. From outside inward: DNA strands reverse and forward; contigs; GC content; GC skew.

strain was slightly higher. On the other hand, both marine strains (*Streptomyces* sp. H-KF8 and *Streptomyces* sp. TP-A0598) showed higher number of genes related to categories of post-translational modification, protein turnover and chaperone functions, as well as in secondary metabolism and translation categories.

Secondary metabolism category comprises 2.3% of the *Streptomyces* sp. H-KF8 genome, being slightly higher when compared to both strains, the soil-derived *S. violaceoruber* A3(2), and the marine-derived *Streptomyces sp.* TP-A0598, accounting for 1.9% and 2.0% of their genomes, respectively. A bioinformatics analysis was performed using the antiSMASH tool to detect biosynthetic gene clusters (BGCs) present in *Streptomyces* sp. H-KF8 that may explain the antimicrobial activity observed, and a total of 26 BGCs were detected (*Undabarrena et al., 2017*). In this report, we show that the spatial distribution of the 26 BGCs are evenly allocated throughout the contigs of *Streptomyces* sp. H-KF8 genome (Fig. 5), which were grouped into 11 different types (NRPS, PKS, hybrids, terpenes, RiPP, ectoine, melanine, siderophores, lantipeptides and butyrolactones). Furthermore, a comparison of the BGCs present in *Streptomyces* sp. H-KF8 with other known BGCs deposited into the MIBiG database, was performed (Table 3). In this line, *Streptomyces* sp. H-KF8 bears two NRPSs BGCs with very low similarity to BGCs involved in the synthesis of the lipoglycopeptide antibiotic mannopeptimycin, produced by *S. hygroscopicus* (*Magarvey et al., 2006*); and the streptolydigin antibiotic, which interferes with the RNA elongation

**Table 3  Biosynthetic gene clusters (BGCs) for secondary metabolites in *Streptomyces* sp. H-KF8.**

| AntiSMASH type descriptor | Scaffold | Length (bp) | Predicted product (%[a]) | MIBiG-ID |
|---|---|---|---|---|
| NRPS | tig_02 | 81,285 | Streptolydigin (13%) | BGC0001046 |
| NRPS | tig_02 | 79,174 | Mannopeptimycin (7%) | BGC0000388 |
| PKS T1 | tig_138 | 33,925 | Kirromycin (6%) | BGC0001070 |
| PKS T2 | tig_139 | 42,512 | Spore Pigment (83%) | BGC0000271 |
| NRPS-PKS T1 | tig_138 | 50,808 | SGR PTMS (100%) | BGC0001043 |
| NRPS-PKS T1 | tig_139 | 52,764 | Neomycin (5%) | BGC0000710 |
| NRPS-PKS T1 | tig_02 | 56,103 | Himastatin (12%) | BGC0001117 |
| NRPS-PKS T3 | tig_02 | 54,318 | Furaquinocin A (21%) | BGC0001078 |
| Terpene-Siderophore | tig_02 | 50,603 | Isorenieratene (100%) | BGC0000664 |
| Nucleoside-Phosphoglycolipid | tig_00 | 35,469 | Moenomycin (100%) | BGC0000805 |
| Oligosaccharide-PKS T1 | tig_16 | 42,574 | Stambomycin (52%) | BGC0000151 |
| Lantipeptide-PKS T1 | tig_138 | 61,004 | Unknown | – |
| Terpene | tig_02 | 26,858 | Hopene (76%) | BGC0000663 |
| Terpene | tig_00 | 20,992 | Unknown | – |
| Terpene | tig_02 | 21,253 | Unknown | – |
| Terpene | tig_02 | 22,162 | Unknown | – |
| Terpene | tig_138 | 21,220 | Albaflavenone (100%) | BGC0000660 |
| Lantipeptide | tig_02 | 21,819 | Unknown | – |
| Lantipeptide | tig_139 | 24,585 | Unknown | – |
| Bacteriocin | tig_02 | 11,412 | Unknown | – |
| Lassopeptide | tig_10 | 22,692 | Unknown | – |
| Siderophore | tig_139 | 11,808 | Desferrioxiamine B (83%) | BGC0000940 |
| Butyrolactone | tig_14 | 11,073 | Griseoviridin/Viridogrisein (11%) | BGC0000459 |
| Ectoine | tig_139 | 10,398 | Ectoine (100%) | BGC0000853 |
| Melanin | tig_139 | 10,509 | Melanin (100%) | BGC0000910 |
| Other | tig_00 | 43,290 | Stenothricin (13%) | BGC0000431 |

**Notes.**
[a]Percentage of genes from known BGCs that show similarity to genes predicted for BGCs from *Streptomyces* sp. H-KF8.

by inhibition of the bacterial RNA polymerase (*Olano et al., 2009*), with 7% and 13% of gene similarity, respectively (Table 3). The two PKSs predicted in *Streptomyces* sp. H-KF8 genome corresponds to the type II spore pigment BGC showing 83% of gene similarity, and also another BGC where only 6% of gene similarity to the antibacterial kirromycin BGC from *S. collinus* Tü 365 was found (*Weber et al., 2008*) (Table 3). A total of eight hybrid clusters, where four of them are PKS-NRPS hybrids were also predicted, which presented low gene similarities with other known BGCs, except for one NRPS-PKS type I cluster (Table 3). In addition, other BGCs found in *Streptomyces* sp. KF8 included five terpenes BGCs, two lantipeptides and two ribosomally synthesized and post-translationally modified peptides (RiPPs) such as the lassopeptide and bacteriocin BGCs. In general, only six BGCs from *Streptomyces* sp. H-KF8 genome displayed 100% gene similarity to their most related known cluster. Examples of these consists on the BGC for the previously mentioned antibiotics moenomycin (*Wallhausser et al., 1965*; *Ostash, Saghatelian & Walker, 2007*) and albaflavenone (*Zhao et al., 2008*) (Table 3). Additionally, BGCs for the aromatic carotene
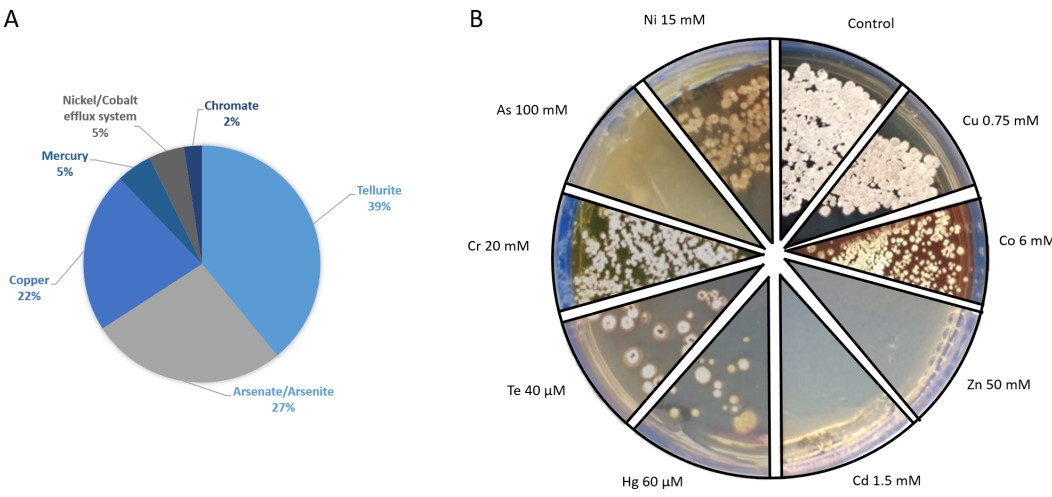

**Figure 6** **Metal-tolerance response in *Streptomyces* sp. H-KF8.** (A) Genetic determinants involved in metal-resistance observed by genome mining. (B) Functional response of metal-resistance in TSA-ASW agar plates. Images show maximum concentration of metal(loids) where growth of *Streptomyces* sp. H-KF8 was observed. Concentrations below these values also presented growth. Control, agar plate without any metal.

isorenieratene, involved in anoxigenic photosynthesis in *S. griseus* (*Krügel et al., 1999*), the conserved osmolite ectoine, that may provide protection from osmotic stress (*Prabhu et al., 2004*; *Graf et al., 2008*) and the melanin pigment clusters (*Guo et al., 2014*; *Sivaperumal, Kamala & Rajaram, 2015*) were observed with 100% similarity. Most of the BGCs (65%) presented low similarity to BGCs of known compounds, evidencing the potential of *Streptomyces* sp. H-KF8 strain to produce novel bioactive molecules.

Due to the dynamics of environmental parameters from the unique isolation site of *Streptomyces* sp. H-KF8, genome mining of pathways involved in response to abiotic stressors such as heavy metals, oxidative stress and antibiotics were also analyzed in this study, in order to unveil genetic determinants that may explain tolerance to these stressful environmental conditions.

## Functional response to heavy metals and metalloids

Genetic determinants involved in heavy metal-resistance in *Streptomyces* sp. H-KF8 were analyzed by genome mining, and at least 49 predicted genes may be playing a role in such tolerance (Fig. 6A). Amongst these, the most abundant genes were related to tellurite, followed by arsenate, copper and mercury, and, to a lesser extent, chromate, nickel and cobalt tolerance (Fig. 6A). Tellurite resistance genetic determinants involved seven *terD* genes, four *terB*, two *yceC* genes, one *terC* gene and one *tehB* gene that encodes a tellurite methyltransferase. In addition, 11 genetic determinants for arsenic tolerance were found, involving three *arsC* genes encoding arsenate reductases, two genes *arsA* encoding arsenical pump-driving ATPases, five genes *arsR* encoding arsenical transcriptional regulators, and the arsenical resistance protein encoding gene *acr3*. Genetic determinants encoding for copper resistance genes, included *copA* and *mco* genes encoding multicopper oxidases, *copD* encoding a copper resistance protein, two genes *ycnJ* encoding

for copper transport proteins, and two genes for the copper-sensing transcriptional regulator, *csoR*. Mercury resistance genes consisted in the mercury reductase encoding gene *merA,* and the mercury transcriptional regulator *merR*. In addition, the *czcD* and *rcnA* genes coding for efflux pumps for cadmium, zinc, cobalt and nickel, respectively, together with the *chrR* gene encoding a chromate reductase, and general heavy metal tolerance such as the *hmt1* gene and seven genes encoding for *merR*-family transcriptional regulators, were also found. Considering all the genetic determinants listed above, we attempted to determine if *Streptomyces* sp. H-KF8 was able to grow on various metal-containing media. *Streptomyces* sp. H-KF8 was able to tolerate copper-, cobalt-, mercury-, tellurite-, chromate- and nickel-containing media, as shown in Fig. 6B for the maximum concentrations tested. Despite the arsenic tolerance-related genes present in *Streptomyces* sp. H-KF8 genome, comprising 27% of the total number of metal-related genes, no evident growth of *Streptomyces* sp. H-KF8 was perceived in this metalloid-containing medium, even in the two different toxic forms of arsenic tested: arsenate and arsenite. Also, no growth was observed in media containing cadmium or zinc.

## Functional response to oxidative stress

A significant amount of genes (69 genes) that may participate in the detoxification of reactive oxygen species (ROS) were found within the *Streptomyces* sp. H-KF8 genome (Fig. 7A). Genes for mycothiol biosynthesis (20 genes), thioredoxin and thioredoxin reductases system (11 genes), alkyl hydroperoxide reductases (nine genes), glutaredoxin and glutathione peroxidase system (four genes), catalases (three genes), and superoxide dismutases (three genes), among others, were identified (Fig. 7A). Interestingly, genes involved in osmotic stress detoxification of chlorinated and brominated compounds such as three *bpo* genes encoding for bromoperoxidases, one *cpo* gene encoding for a chloroperoxidase and one gene encoding for a chlorite dismutase were also present in *Streptomyces* sp. H-KF8 genome (Fig. 7A). Concerning transcriptional regulators controlling the redox balance, transcriptional factors from *perR*, *rex*, *lysR* and *soxR* families, were also present. Due to an important genetic content of oxidative stress related genes, response of *Streptomyces* sp. H-KF8 to the toxic $H_2O_2$ was tested, and compared to the model streptomycete *S. violaceoruber* A3(2). At various $H_2O_2$ concentrations, *Streptomyces* sp. H-KF8 displayed smaller susceptibility areas against the toxic, in comparison with *S. violaceoruber* A3(2) (Figs. 7B and 7C, respectively). A significant difference of the susceptibility areas among the two strains was observed at concentrations of 1 M, 2 M and 4 M of $H_2O_2$, indicating a major resistance response of *Streptomyces* sp. H-KF8 towards $H_2O_2$ toxicity (Fig. 7D).

## Functional response to antibiotics

Antibiotic-producing *Streptomyces* strains usually encode resistance genes within their BGCs to protect themselves against the noxious action of the synthetized compound (Zotchev, 2014). In this line, resistance of *Streptomyces* sp. H-KF8 to commercial antibiotics with different biological targets was explored. Genome mining revealed more than 90 genes that could be involved in antibiotic resistance. The most abundant genes

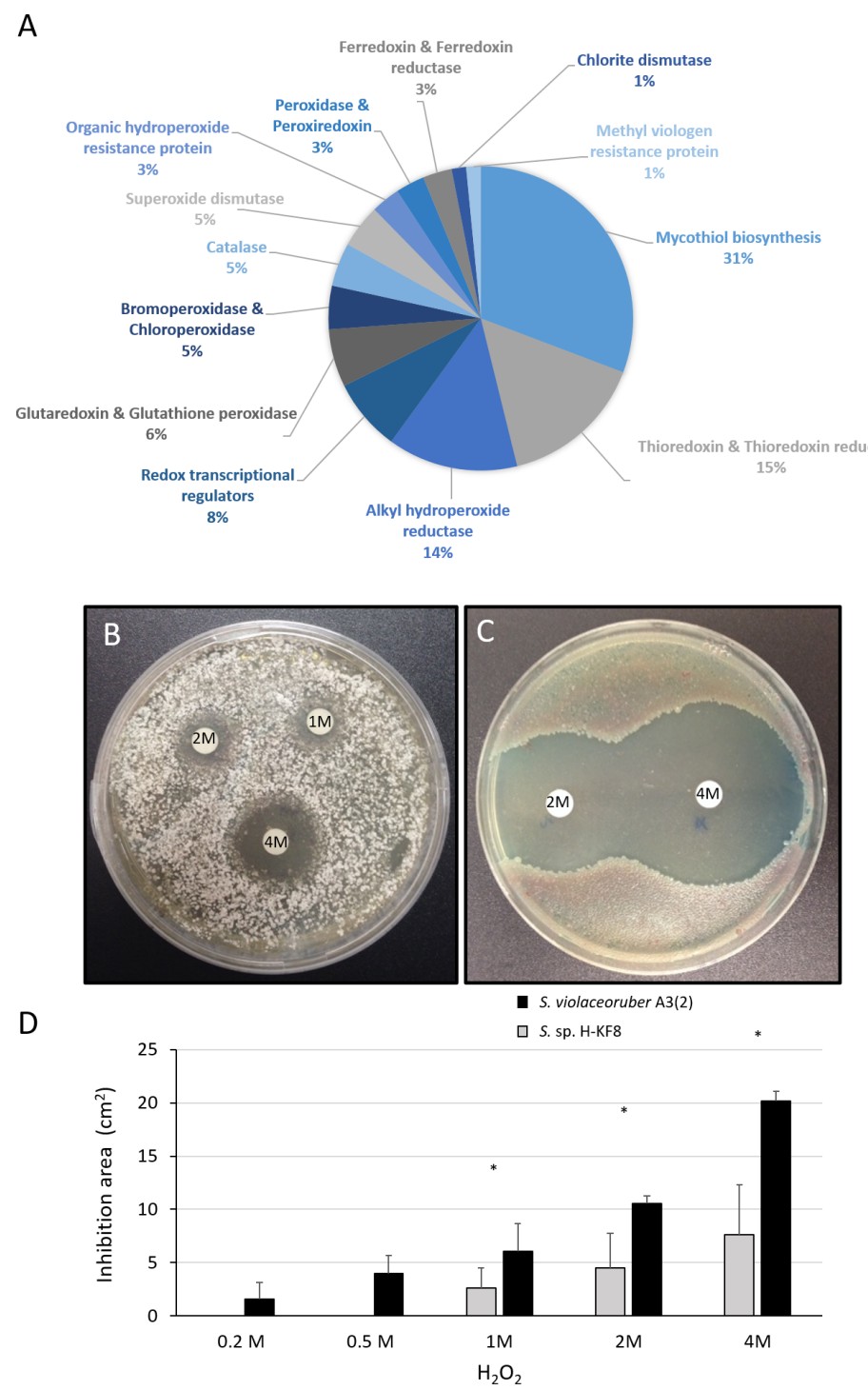

**Figure 7** **Oxidative stress response of *Streptomyces* sp. H-KF8.** (A) Genetic determinants involved in oxidative stress-resistance observed by genome mining. Functional response of (B) *Streptomyces* sp. H-KF8 and (C) *Streptomyces violaceoruber* A3(2) respectively, showing comparative inhibition zones with hydrogen peroxide where the concentration of hydrogen peroxide used in each disk is shown. (D) Quantitative assay of inhibition area of both *Streptomyces* strains facing several concentrations of hydrogen peroxide. Asterisks indicate significant differences between strains ($t$-Test considering a $p$-value <0.01).

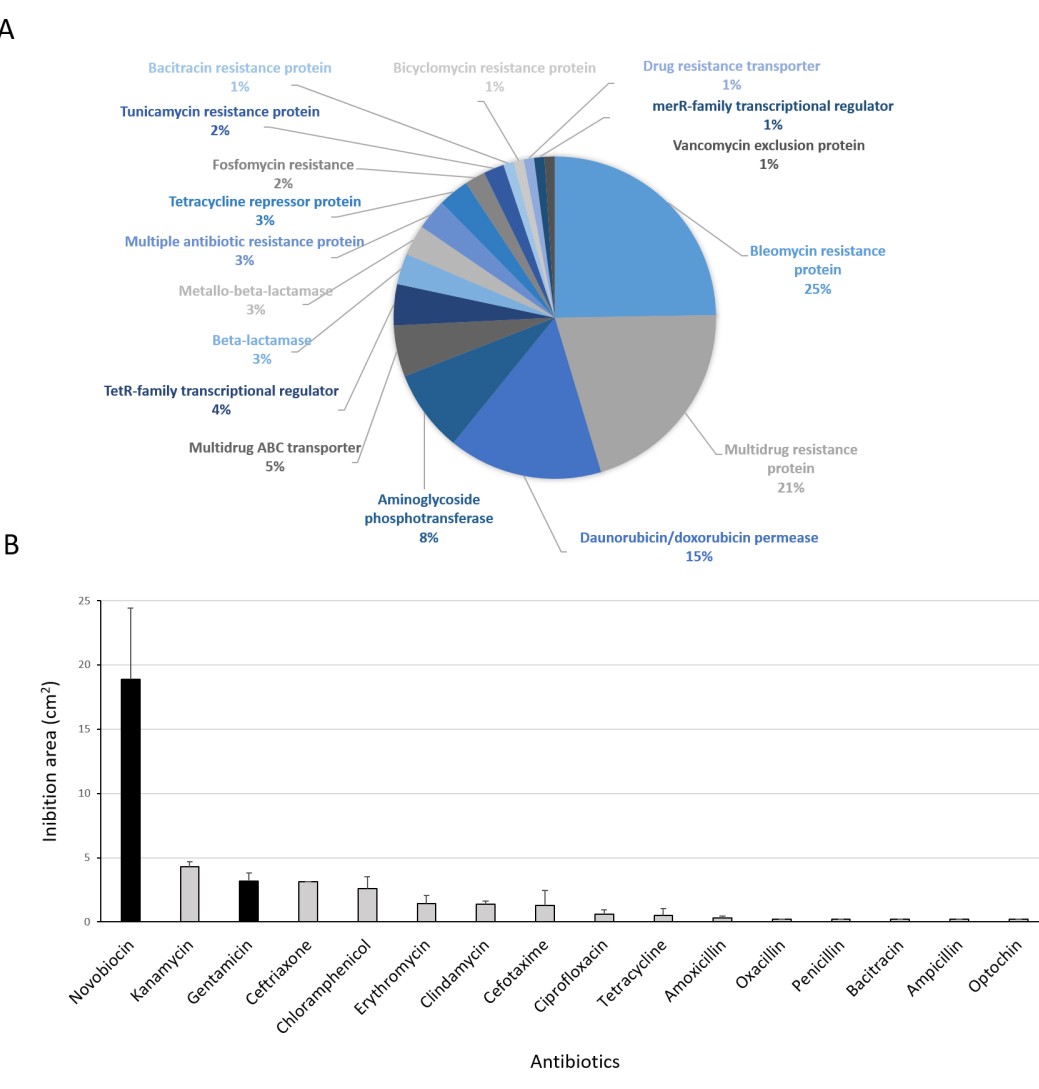

**Figure 8 Antibiotic-resistance response in *Streptomyces* sp. H-KF8.** (A) Genetic determinants involved in antibiotic-resistance observed by genome mining. (B) Functional response of antibiotic-resistance in MH-ASW agar plates. Black columns indicate susceptibility to the antibiotic tested and grey columns indicate resistance to the antibiotic tested.

encode for bleomycin resistance proteins (24 genes). Specific resistance genes related to modification and inactivation of antibiotics such as aminoglycoside phosphotransferases (eight genes), $\beta$-lactamases (three genes), metallo-$\beta$-lactamases (three genes), and one gene for erythromycin esterase and penicillin amidase, respectively, were identified (Fig. 8A). In addition, genes for efflux of toxic compounds including multidrug resistance proteins (20 genes), daunorubicin/doxorubicin ABC transporter permeases (15 genes), multidrug ABC transporters (seven genes) and one gene encoding for a multidrug MFS transporter, were detected (Fig. 8A). Among the transcriptional regulators, the TetR-family transcriptional regulators were the most abundant, with 10 genes. Also, the marR-family transcriptional regulator and three *marR* genes encoding for multiple antibiotic resistance proteins were identified (Fig. 8A). In the functional assay against 16 different antibiotics

tested, *Streptomyces* sp. H-KF8 exhibited an 88 % of resistance-response, being susceptible to only two antibiotics: novobiocin, which targets the DNA gyrase, and gentamicin, which inhibits protein synthesis by irreversibly binding to the 30S subunit of the bacterial ribosome (Fig. 8B).

## DISCUSSION

In this report, phenotypic analysis of *Streptomyces* sp. H-KF8 in several agar media was assessed, revealing in general one week of incubation time to obtain colonies and two weeks for sporulation; although growth rates, sporulation rates and pigmentation differs throughout the different media used. Antimicrobial production in *Streptomyces* sp. H-KF8 was enhanced in late growth phase (>10 days) and favoured in media where sporulation was observed. *Streptomyces* genus is characterized for slow growth and a complex developmental life cycle (*Flärdh & Buttner, 2009*). Physiological differentiation is tightly linked to secondary metabolism and hence, sporulation capacities of *Streptomyces* might enhance the discovery of new compounds (*Chater, 2013*; *Kalan et al., 2013*; *Zhu et al., 2015*). In addition, antibiotics synthesis is regulated by environmental nutrients, such as carbon sources. Media carbon source has an important effect on antibiotic production, being demonstrated that when bacteria are grown with a preferred carbon source, secondary metabolism seems repressed (*Sánchez et al., 2010*). This may explain the differences in inhibition patterns observed for the *Streptomyces* sp. H-KF8 antagonistic assays displayed in the various media tested, showing a maximum inhibition halo against *S. aureus* after 15 days of incubation. Due to the interesting antibacterial activity of *Streptomyces* sp. H-KF8, its whole-genome was sequenced and previously reported (*Undabarrena et al., 2017*). Thus, in this study an extended genome analysis for *Streptomyces* sp. H-KF8 was performed, in order to gain insights into the mechanisms by which it displays antibiotic biosynthesis and resistance to multiple stressors.

Genome mining has been used in various fields to describe the exploitation of genomic information for the discovery of new processes, targets and products (*Challis, 2008*). Through genome sequencing and bioinformatic analysis using antiSMASH platform (*Medema et al., 2011*; *Blin et al., 2013*; *Weber et al., 2015*), it is possible to address the secondary metabolic potential of a strain by identification of its biosynthesis gene clusters (BGCs) (*Iftime et al., 2016*). A total of 26 BGCs were previously detected in *Streptomyces* sp. H-KF8 genome (*Undabarrena et al., 2017*). In this report, an extended genetic analysis including the distribution of these BGCs along *Streptomyces* sp. H-KF8 genome was determined and comparison with known BGCs from the Minimum Information about a Biosynthetic Gene cluster (MIBiG) database, which compiles a total of 1,170 experimentally characterized known gene clusters (*Medema et al., 2015*) was aimed. *Streptomyces* sp. H-KF8 BGCs include two PKSs, two NRPSs and four hybrid PKS-NRPS, four other hybrids, five terpenes, two lantipeptides, one bacteriocin, lassopeptide, siderophore, butyrolactone, ectoine, melanin, and one with unknown classification. Notably, *Streptomyces* sp. H-KF8 presented only six BGCs with 100% similarity to a known cluster; suggesting that most secondary metabolites produced by *Streptomyces* sp. H-KF8 are yet to be elucidated,

and can contribute to the discovery of novel natural products. In this context, genome mining has proven to be a fundamental tool for genome-based natural product discovery (*Jensen et al., 2014*), and has guided the discovery of novel natural products from several marine actinobacteria (*Gulder & Moore, 2010*; *Tang et al., 2015b*). Among these are the aromatic polyketide angucyclinone antibiotic (*Zhang et al., 2012*) and polyene macrolides with antifungal activity (*Tang et al., 2015a*). Moreover, marine *Streptomyces* metabolites are produced by different metabolic pathways in comparison to their terrestrial counterparts (*Li et al., 2011*; *Lee et al., 2014*; *Barakat & Beltagy, 2015*). These metabolites emerge as a result of the unique and dynamic conditions of the ocean, such as high hydrostatic pressure, low temperature, variation in salinity, and depletion of micronutrients proper of the marine environment (*Das, Lyla & Khan, 2006*; *Lam, 2006*; *De Carvalho & Fernandes, 2010*). Despite that marine adaptations are scarcely studied, recent comparative genomics of marine-derived *Streptomyces* unveiled an enrichment in TrK and BCCT transporters, along with the observation that their genomes are generally smaller in size and have a slightly higher GC content in comparison to *Streptomyces* from other environmental sources (*Tian et al., 2016*). *Streptomyces* sp. H-KF8 genome is consistent with these findings, holding distinctive biological and genomic signatures acknowledged for marine *Streptomyces* strains. Therefore, its metabolite biosynthesis may be under marine abiotic selective pressures, hence modulating secondary metabolism production.

Comparative genomics encompassing completely sequenced *Streptomyces* obtained from several isolation sources revealed that the most abundant COG categories were transcription (K), followed by carbohydrate metabolism (G) and amino acid metabolism (E) (*Kim et al., 2015*). This is in agreement with the most abundant categories found in the *Streptomyces* sp. H-KF8 genome, which also could explain the versatility of *Streptomyces* sp. H-KF8 to grow in several media with different carbon sources. Furthermore, in marine-derived *Streptomyces*, a higher proportion of genes belonging to the COG categories of translation (J) and post-translational modification, protein turnover and chaperones (O) was observed (*Tian et al., 2016*). Accordingly, the (J) and (O) COGs categories were also overrepresented in both marine strains analyzed, *Streptomyces* sp. H-KF8 and *Streptomyces* sp. TP-A0598 (*Komaki et al., 2015*), in comparison to the terrestrial *Streptomyces violaceoruber* A3(2) (*Bentley et al., 2002*). This may indicate an important role of protein metabolism in marine environments, probably due to the active responses against abiotic stressors and the dynamics that microorganisms have to overcome to survive in these extreme ecosystems. In addition, our analysis showed an increase in the categories of cell cycle control, cell division, chromosome portioning (D), secondary metabolism (Q) and defense mechanisms (V), for both marine strains in comparison to *Streptomyces violaceoruber* A3(2). Percentage of the COG category for defense mechanisms (V) in *Streptomyces* sp. H-KF8 was interestingly higher (2,81%) than in *Streptomyces* sp. TP-A0598 (1,8%), and comparatively similar with what was observed for deep-sea bacteria (3,0%) (*Qin et al., 2011*). As the defense mechanism category includes genes for resistance to heavy metals, osmotic and oxidative stress as well as antibiotics, the functionality of these biological traits was evaluated for *Streptomyces* sp. H-KF8, and notably, an important resistance to these multiple stressors was evidenced.

Environmental pollution by heavy metals can arise due to anthropogenic and/or geogenic sources. Although metal-resistant strains isolated from contaminated areas have been described (*Amoroso et al., 2001*; *Schmidt et al., 2005*; *Schmidt et al., 2009*; *Polti, Amoroso & Abate, 2007*; *Albarracin et al., 2008*; *Haferburg et al., 2008*; *Siñeriz, Kothe & Abate, 2009*; *Lin et al., 2011*; *El Baz et al., 2015*), there is limited information about the physiology of *Streptomyces* in presence of environmental metal pollutants. Due to the naturally high concentrations of certain heavy metals in Chilean northern Patagonia (*Guevara et al., 2004*; *Revenga et al., 2012*; *Hermanns & Biester, 2013*) product of the highly active seismic and volcanic activity (*Pantoja, Luis Iriarte & Daneri, 2011*), the ability of *Streptomyces* sp. H-KF8 to grow in several metal(loid)s supplemented media was evaluated. Surprisingly, resistance to copper, cobalt, mercury, tellurite, chromate and nickel was revealed.

Interestingly, the most abundant genes in *Streptomyces* sp. H-KF8 were related to tellurite resistance, involving the tellurite methyltransferase (encoded by *tehB*) and several tellurite resistance genes (*terB*, *terC*, *terD*, *yceC*). Although the *ter* operon has been described previously (*Taylor, 1999*), specification of its mechanism of action remains obscure (*Chasteen et al., 2009*). Mainly, it has been shown that tellurite detoxification is via enzymatic reduction by several flavoprotein-mediated non-specific metabolic enzymes (*Arenas-Salinas et al., 2016*), or by non-enzymatic mechanisms mediated by intracellular thiols like glutathione (*Turner et al., 2001*). Either way, tellurite reduction generates oxygen reactive species (ROS), especially superoxide anion ($O_2^-$), which is deleterious to fundamental cell macromolecules producing protein oxidation, lipid peroxidation and DNA damage (*Pérez et al., 2007*; *Tremaroli, Fedi & Zannoni, 2007*). Surprisingly, *Streptomyces* sp. H-KF8 did not show black pigmentation after tellurite exposure, which is a distinctive phenotype that indicates tellurite reduction to elemental tellurium (*Taylor, 1999*), suggesting that other mechanisms of resistance could be involved in *Streptomyces* sp. H-KF8. To our knowledge, this is the first tellurite-resistant *Streptomyces* strain described so far.

Additionally, resistance to mercury at a concentration of 60 μM was observed for *Streptomyces* sp. H-KF8. In general, bacteria capable of resisting mercury above 20 μM, should possess specific detoxification systems, as mercury is one of the most toxic elements on earth and produces several health concerns for macroorganisms (*Das, Dash & Chakraborty, 2016*). In bacteria, two different resistance operons are known, the basic narrow-spectrum *mer* operon *merRTPA* for inorganic mercury, and the broad-spectrum operon that additionally contains *merB*, which provides protection against organomercurial compounds (*Barkay, Miller & Summers, 2003*). In addition, it was recently demonstrated that mercury resistance mechanisms could also be involved in tellurite cross-resistance (*Rodríguez-Rojas et al., 2015*). Studies in *Streptomyces* includes *S. lividans* 132, that carries two divergently transcribed operons named *merAB* and *merRTP* in the chromosome (*Sedlmeier & Altenbuchner, 1992*; *Brünker et al., 1996*; *Rother, Mattes & Altenbuchner, 1999*), and two *Streptomyces* spp. strains isolated from estuarine sediments where these genes were also observed in giant linear plasmids (*Ravel, Schrempf & Hill, 1998*; *Ravel et al., 2000*). Interestingly, the genetic operons mentioned above were not detected in *Streptomyces* sp. H-KF8, despite the fact that a mercury-resistance phenotype was evidenced. Instead, the presence of two mercury-related genes, the transcriptional

regulator *merR* and the mercuric reductase *merA*, may be playing a role in such resistance. MerA is a flavoprotein NADPH-dependent enzyme responsible for the reduction of mercury(II) to the elemental and less toxic volatile mercury(0) (*Barkay, Miller & Summers, 2003*). Similarly, evidence of functional operons conformed either by *merA* or *merRA* have been previously reported in archaea (*Boyd & Barkay, 2012*).

However, no evident growth was observed in the presence of arsenate or arsenite, although *Streptomyces* sp. H-KF8 bears at least 11 genetic determinants that could involved in its detoxification. In general, the arsenic resistance operon consists of *arsRABCD* genes, where *arsC* encodes for an arsenate reductase that converts arsenate to arsenite, which is then exported through the ArsAB ATPase-efflux pump. In *Streptomyces* sp. H-KF8, *arsA*, *arsC* and *arsR* genes are present, but lack the *arsB* gene, which encodes an arsenite antiporter, crucial for anchoring ArsA to the inner membrane with concomitant detoxification of arsenite. Absence of the *arsB* gene may explain the sensitivity of *Streptomyces* sp. H-KF8 towards these toxics. Arsenic resistance genes are generally widespread amongst both Gram-positive and Gram-negative bacteria, reflecting its broad distribution in the environment (*Silver & Phung, 2005*). In fact, these genes were also conserved in several marine streptomycetes from the South China Sea (*Tian et al., 2016*).

*Streptomyces* sp. H-KF8 displayed a notorious copper-resistant phenotype, concordant with the detection of three *copA* genes encoding for multicopper oxidases that may be responsible for the oxidation of Cu(I) to its less toxic form Cu(II) (*Hobman & Crossman, 2014*). Copper is an essential metal for living beings, but is extremely toxic at higher concentrations (*Gaetke & Chow, 2003*). Moreover, Chile is the major copper-producing country in the world, due to its geological nature (*Wacaster, 2015*). Hence, the widespread of copper resistant genetic determinants that has been demonstrated in Chilean marine sediments (*Besaury et al., 2013*) is expected.

Resistance to nickel and cobalt in *Streptomyces* sp. H-KF8 might be given by the *rcnA* gene that participates in the efflux system of these metals. Highly nickel- and cobalt-resistant *Streptomyces* were found in an acid mine drainage, where growth in media containing up to 10 mM nickel(II) or 3 mM cobalt(II) was observed (*Schmidt et al., 2005*). In this report, *Streptomyces* sp. H-KF8 was able to grow even at higher concentrations: 15 mM nickel(II) and 6 mM cobalt(II), respectively. Furthermore, chromate toxicity (20 mM) might be overcome in *Streptomyces* sp. H-KF8 due to the presence of the *chrR* gene encoding a chromate reductase involved in the enzymatic reduction of chromate to the less harmful chromite cation (*Das, Dash & Chakraborty, 2016*). Previously reported *Streptomyces* chromate-resistant strains isolated from sugar cane plant were able to grow in 17 mM, where also chromate-removing activity was demonstrated (*Polti, Amoroso & Abate, 2007*).

Metal exposure and adverse abiotic environmental factors produces a general condition of oxidative stress in microorganisms. As oxidative stress is hazardous for fundamental macromolecules, bacteria have evolved several mechanisms to protect themselves from these environmental stresses. In *Streptomyces* sp. H-KF8, an exceptional response to several concentrations of $H_2O_2$ was observed, compared to the model *Streptomyces violaceoruber* A3(2) which was more susceptible towards the toxic. Consequently, a wide number

of genetic determinants related to ROS response were present in the *Streptomyces* sp. H-KF8 genome. Remarkably, a high number of thioredoxins (*trx*) and alkyl hydroperoxide reductases (*ahp*) genes (nine of each) were found in *Streptomyces* sp. H-KF8, in comparison with *Streptomyces violaceoruber* A3(2) where five and one genes were described, respectively. The *ahp* and *trx* are fundamental $H_2O_2$-inducible genes that encodes for enzymes known to participate in the bacterial response to oxidative stress, which are regulated by *oxyR* in *E. coli* (*Storz & Imlay, 1999*; *Seaver & Imlay, 2001*; *Chiang & Schellhorn, 2012*). The *oxyR* regulon is not present in *Streptomyces* sp. H-KF8, but instead two copies of the *perR* regulator fulfill its role in Gram-positive bacteria (*Ricci et al., 2002*; *Dubbs & Mongkolsuk, 2012*). Also, the *ohrR* transcriptional regulator that senses organic peroxide (ROOH) and sodium hypochlorite (NaOCl) (*Dubbs & Mongkolsuk, 2012*) was found in *Streptomyces* sp. H-KF8. In addition, several genes regulated by the *soxR* transcriptional regulatory system such as glutaredoxin and glutathione peroxidase, superoxide dismutases (*sod*), catalases (*kat*) and thioredoxin reductases were recognized in *Streptomyces* sp. H-KF8 genome, which overall may be accounting for it resistance through $H_2O_2$ exposure. Even more, the chromate reductase (*chrR*) previously mentioned, could also provide additional protection against $H_2O_2$ (*Das, Dash & Chakraborty, 2016*). Interestingly, unusual genes encoding for bromoperoxidases, chloroperoxidases and chlorite dismutases, involved in osmotic stress detoxification of brominated and chlorinated toxic compounds which are abundant in the marine environments (*Sander et al., 2003*; *Bouwman et al., 2012*), were also present in *Streptomyces* sp. H-KF8 genome. On the other hand, *Streptomyces violaceoruber* A3(2) possess only one chloroperoxidase, suggesting that this might represent another marine adaptation trait for *Streptomyces* sp. H-KF8. Osmotic and oxidative stress response seems to be regulated via a network of sigma factors in *Streptomyces violaceoruber* A3(2), that controls the activation of several oxidative defense proteins, chaperones and systems that provide osmolytes and mycothiol (*Lee et al., 2005*). Consistently, a high amount of genes for mycothiol biosynthesis was identified in *Streptomyces* sp. H-KF8. Mycothiol is the major low-molecular-weight thiol present in actinobacteria, and serves as a buffer to advert disulfide stress, in complement of the enzymatic system presented above (*Buchmeier & Fahey, 2006*; *Den Hengst & Buttner, 2008*).

Recently, evidence of heavy metal driving co-selection of antibiotic resistance in both natural environments (*Seiler & Berendonk, 2012*) and contaminated ones (*Li, Li & Zhang, 2015*; *Henriques et al., 2016*) have been reported. In this line, isolation of *Streptomyces* with both metal and antibiotic co-resistances have been described (*Van Nostrand et al., 2007*). In addition, co-evolution of resistance within closely related antibiotic-producing bacteria has been demonstrated for *Streptomyces* (*Laskaris et al., 2010*). Hence, the antibiotic response against pharmaceutical compounds was investigated in *Streptomyces* sp. H-KF8, and resistance was observed to all antibiotics tested, with exception of gentamicin and novobiocin. Resistance to almost all antibiotics tested, could be due to the presence of multiple BGCs with different mode of action. A typical BGC cluster that produces a bioactive compound is generally coupled to its corresponding resistance gene (*Zotchev, 2014*). The phenomena of widespread distribution antibiotic resistance genes in natural environments is consequence of improper use of antibiotics in medical treatment,

as well as by an indiscriminate use in agriculture, livestock and aquaculture (*Brown et al., 2006*). Phenomena such as the grasshopper effect may also contribute to the rapid transport of toxics around the globe through atmospheric and oceanic currents (*Sadler & Connell, 2012*).

Overall, our study shows the response of a marine *Streptomyces* sp. H-KF8 against several abiotic stressors such as heavy metals, oxidative stress and antibiotics, along with the genome mining of the biosynthetic gene clusters that could be involved in the antimicrobial activity observed. Altogether, these biological features may enable *Streptomyces* sp. H-KF8 to thrive in the complex fjord marine environment.

### Funding

This work was supported by the "Comisión Nacional de Investigación Científica y Tecnológica" (FONDECYT 11121571) to BC and Swedish Research Council No 2013-6713. MS was supported by FONDECYT 1151174 and JU was supported by FONDECYT 11140666. AU was supported by Conicyt PhD fellowship and Conicyt Gastos Operacionales (No 21120621). The funders had no role in study design, data collection and analysis, decision to publish, or preparation of the manuscript.

### Grant Disclosures

The following grant information was disclosed by the authors:
Comisión Nacional de Investigación Científica y Tecnológica.
FONDECYT: 11121571, 1151174, 11140666.
Swedish Research Council: No 2013-6713.
Conicyt Gastos Operacionales: 21120621.

### Competing Interests

The authors declare there are no competing interests.

### Author Contributions

- Agustina Undabarrena conceived and designed the experiments, performed the experiments, analyzed the data, wrote the paper, prepared figures and/or tables, reviewed drafts of the paper.
- Juan A. Ugalde analyzed the data, contributed reagents/materials/analysis tools, reviewed drafts of the paper.
- Michael Seeger analyzed the data, reviewed drafts of the paper.
- Beatriz Cámara conceived and designed the experiments, analyzed the data, contributed reagents/materials/analysis tools, wrote the paper, reviewed drafts of the paper.

### Data Availability

NCBI GenBank accession number of the whole-genome shotgun project LWAB00000000 http://www.ncbi.nlm.nih.gov/bioproject/317393.

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
