# Peer review of "­Genomic data mining of the marine actinobacteria Streptomyces sp. H-KF8 unveils insights into multi-stress related genes and metabolic pathways involved in antimicrobial synthesis"

_PeerJ, doi:10.7717/peerj.2912_

## Round 0.1 · original submission · Major Revisions

This is an interesting case where 3 referees have very different opinions. however, due to the valid concerns of reviewer number 3 that obviously read very carefully the ms, I strongly suggest to acknowledge all the issues and explain better in the text. However, I do not agree with reviewer 3 about the english, the english is good, is just that you need to explain better.

Reviewer 1 ·

Basic reporting

The manuscript is well written, figures are good. However, the literature review throughout the text is very extensive and exhaustive.

Experimental design

Enough details are presented.

Validity of the findings

Well done.

Additional comments

The manuscript is well written, informative and massively discussed. However, it is very extensive (may be reduced).
Despite the quality of manuscript, I am still concerning about the possibility of superposed content with the one presented in reference “Undabarrena A., Ugalde JA, Castro-Nallar E., Seeger M., Cámara BP. 2016b. Nearly-complete genome sequence of Streptomyces sp. strain H-KF8 a marine bacterium exhibiting antibacterial activity isolated from a northern Chilean Patagonian Fjord. Under revision”. Did authors confirm no superposed contents?

·

Basic reporting

Current paper presents an interesting data set on genomic data mining of the marine actinomycete Streptomyces sp. H-KF8 unveils insights into multi-stress related genes and metabolic pathways involved in antimicrobial synthesis. The manuscript is well written and should be in focus of classical microbiology, biotechnology and molecular biology. In general the manuscript presented in an intelligible fashion and well written in standard English. Overall this manuscript presents the data well, and I recommend that it can be published after some minor revisions I have suggested below.

Experimental design

–I would suggest to rewrite l. 163-165 due to it’s unclear a qualitatively ranging of inhibition zones. Did you use any standards or references for determinations of inhibition zones?
–It’s unclear – why did you chose this strain and what peculiarities in the fjords microbial consortia and fjords marine environment exist? Please add this explanation in the MS.
–Did you sequence a genome of strain or did you use a data already available in GenBank only? Please, add a history and known/published information about this strain.
–Why did you use another strain (S. violaceoruber A3(2)) for comparison of functional response to oxidative stress but didn’t use this strain for other experiments. It would be good if you have the same set of experiments related to functional response to heavy metals and metalloids of strain Streptomyces violaceoruber A3(2). In this focus the comparative analysis looks no finished.
–Could you please to unify the concentrations of antibiotics in l. 230-233. The concentration of Optochin is missed.

Validity of the findings

The data obtained in this study looks representative and statistically analyzed. As authors postulate, cit. “This study revealed that the marine-derived strain H-KF8 has the capability to tolerate a diverse set of heavy metals such as copper, cobalt, mercury, chromium and nickel; as well as the highly toxic tellurite, a feature first time described for Streptomyces.”

Additional comments

–Could you please to add a size of colonies in l.250.
–Could you please to reduce a discussion section on 10-20 %.
–I would suggest to avoid numerous abbreviations made by authors.
Could you please to use :
-“natural products” instead of “NP” (l.46, 47, 49, 51, 111, etc.);
- full name of studied strain – “Streptomyces sp. H-KF8” instead of “ strain H-KF8 or H-KF8” (l.36, etc.);
- full name of target strains S. aureus instead of STAU , and others- ( for example - l.160-l.165, l.282in results, etc.);
–You used the abbreviation of “LMW” only once. Please delete it.
–Please, unify the name of metal ions (for example – l.200, l.349, l.561 and etc.)
–Please, check a fig. 3. What strain did you use for outgroup? Was it Amycolatopsis?
–Please, add a scale on fig. 4
–It would be good if you add a figures containing the influence of all metals and all concentrations on the studied strain. You also may to add this pictures to fig.7 or move them to the supplementary.
–What does it mean a numbers on the paper disks presented on the fig.8 ? Is it a concentration of H2O2? Please add a legend.
–Could you please to avoid aggressive colors used for figures and graphs, especially fig. 8-9. Please, compare the view of colored signs of graphs and black signs for circular diagrams (fig. 7-9)

Reviewer 3 ·

Basic reporting

I respectfully suggest that the authors get editing help from proofreader with proficiency in scientific English skills to improve manuscript and make it sufficiently comprehensible and clear.

Relevant prior literature IS NOT appropriately referenced.
1. The whole purpose of the manuscript relies on efforts of data-mining, using a previously sequenced genome cited in line 180 as (Undabarrena et al., 2016b). Sequenced genome to which data-mining must be referenced is a manuscript "under revision" as declared in line 180 and 1029. Since this is another term for "not yet published", I made an updated search of the work, unsuccessfully.

2. Regarding literature that serves as framework and background, the manuscript is flawed. Authors fail to accurately justify the selection of the strain H-KF8. Although it is widely accepted that actinobacterial bioprospection efforts at unexplored niches are urgent, it is also important to state criteria for selection of strains to study further. Screening and scrutiny procedures must have been previously done that led authors to conclusions such as "Streptomyces sp. strain H-KF8 ... displays an important antimicrobial activity". This should have been stated or demonstrated or referenced.

Experimental design

Phylogenetic analysis (line 143)
It is hard to understand the reasons to construct a phylogeny based on 16s rRNA only. Also, it is not clear how the evolutionary model was calculated.

Methodology regarding Genome Mining and Bioinformatic analysis (page 9, line 177), is not sufficiently described.
- DNA extraction protocol is not mentioned nor referenced (which is not trivial since we are discussing a marine Streptomyces; this genus is particularly recalcitrant (1).
- Sequencing platform technology is not mentioned, neither described. Although we can assume, based on the software described (lines 179-192), that new generation sequencing or
- Since the sequenced genome is not yet published (referenced as "under revision in line 180), the following information must be provided:
- Reads size, quality as well as adapters used for sequencing purposes (assuming this platform was i.e. Illumina) are not declared.
- Sequencing coverage is unknown from the information provided in this manuscript, which leave.

Antimicrobial activity. Lines 159-166

1. (Line 159). Antimicrobial activity cannot be measured by crossed-streak or double layer methods. These are qualitative methods that allow us rapid screens for antagonisms and later selection, measurements and identification of antimicrobial agent(s). I only suggest these methods when authors need a scrutiny for hundreds of strains, but not to measure activity. This study is also ambiguous since it cannot be compared to previously extensive reported work among genus Streptomyces.

Qualitative work is mostly valuable when experimental design will allow us determine singularity of our data/individual. Nevertheless, this is not the case of the manuscript. Extensive research of antimicrobial activity of Streptomyces strains isolated around the world is being published each month, and several antibiotic producing model organisms belong to this genus, so if qualitative approach is selected, reference strains must be included to make appropriate conclusions about the relevance of the strain. We encourage authors to edit manuscript stating how a previous screening allowed them to select this strain to work with, and reproducibly measure antibiotic activity with short-time consuming methods (Methods for in vitro evaluating antimicrobial activity: A review)

Antibiotic resistance evaluation could need a better experimental design obtaining a Minimal inhibitory concentration, that can be compared to other strains.

1. Sponge-associated actinobacterial diversity: validation of the methods of actinobacterial DNA extraction and optimization of 16S rRNA gene amplification.

Validity of the findings

I find it inappropriate to use as main reference of data source used for this manuscript, an unpublished work, without providing basic information of the genome, and sequencing platform. Methodology for data mining is not complete and unclear, making it difficult to be reproducible.

If the purpose of the phylogenetic "analysis" is to show that sequenced strain belongs to the genus Streptomyces, the phylogenetic tree seems to be enough. However, due to robust data obtained for strain H-KF8 and available in databases, MLSA seems to be the most appropriate method to make results meaningful. Also, calculation methods for the selected evolutionary model are not provided.

Line 302-307. Several comparisons through the manuscript are among Streptomyces violaceoruber and the marine Streptomyces sp. TP-A0598, however, it is not well understood why these strains are selected among the hundreds of reported Streptomyces strains from soil, sea sediments, and other strains that have historically used as models for BGC studies.

Macroscopic and Microscopic Morphologies (Figures 1 and 2) should be better presented. Briefly, macroscopic morphology characterization of Streptomyces species, are standardized through ISP medium are these are not shown. Instead, photographs of H-KF8 strain in other media are shown in Figure 1 and seem to be irrelevant in terms of colony morphology, pigment production, aerial hyphae, etc (which are features standarized for ISP media). Results for this characterization could provide interesting and significant information about the strain that could also be interpreted correlated to genomic information provided in figures 5 and 6. (COGS vs usage of carbon and nitrogen sources).

It is suggested to create a gallery of pictures with zoom to a single colony growing on each ISP medium. Regarding microscopic morphology, Figure 2 is well presented in a SEM photograph, but the light-microscope picture (B) is not sound in terms of microscopic structures. Staining preparations are rapid and effective techniques that will more effectively show spores and mycelia structures.

Regarding phenotypic experiments of strain H-KF8:
Antibiotic activity, response to oxidative stress, resistance to antibiotics and metal-resistance measurements are mostly qualitative.

These results are valid and enough to demonstrate that genome mining is a tool to reveal phenotype, and microbiological test results are consistent with it, but lack novel information and do not allow authors or readers to find relevant information since it cannot be compared to previously extensive reported work among genus Streptomyces due to two main reasons: 1) Experimental design DO NOT INCLUDE reference strains that allow us to compare and determine uniqueness or novelty of the strain. 2) Experimental design is mostly qualitative and only demonstrates properties that are well known for most marine Streptomyces.

Discussion could provide deeper insights; for example, antibiotic resistance pattern of strain H-KF8 could be easily correlated to BGCs, but this point seems to be missed at the end of the discussion section, and authors limit the interpretation to mere possibilities of acquisition of this resistance and missing soundness that could be provided by the genome mining results in this particular feature.

Therefore, this reviewer finds it hard to understand how this work fills the knowledge gap or adds value to the literature in the area.

Additional comments

All of the above are insights in the light of the goals of the authors that were stated in the first lines that the work would provide evidence of a molecular crosstalk response between abiotic stress and how these play an important role in the evolution of secondary metabolism genes. Nonetheless, I encourage authors to consider quantitative methods for phenotypic characterization, and describe them with sufficient information to be reproducible by another investigator. These techniques are not time consuming and require basic accesible equipment. Also, if resubmission of manuscript is considered, I respectfuly suggest deepest discussions of results that enlighten a path to "understand the genetic mechanisms underlying antibiotic biosynthesis and abiotic stress adaptation" (line 116-117).

Annotated reviews are not available for download in order to protect the identity of reviewers who chose to remain anonymous.

---

## Round 0.2 · accepted · Accept

The reviewer who was most critical is now satisfied with all the changes.

Reviewer 3 ·

Basic reporting

No comment

Experimental design

No comment

Validity of the findings

No comment

Additional comments

Line 254:"... and hyphae with Gram staining (Fig 2B)"
- Should be : "... Gram staining (Fig 2C)"